# Development of the advised protocol for OCT study terminology and elements anterior segment OCT extension reporting guidelines (APOSTEL-AS): Study protocol

Ameenat Lola Solebo[1,2,3]*, Marcus Ang[4,5], Alice Bellchambers[1,2,6], Colin J. Chu[7,8], Alastair K. Denniston[9,10], Laura E. Downie[11], Thomas Evans[12], Alexander S. Fraser[13], Scott Hau[7], Alex S. Huang[14], Pearse A. Keane[7,8], Xiaoxuan Liu[9,10,15], Jodhbir S. Mehta[4,16], Giovanni Ometto[6,7,17], Axel Petzold[7,18,19], Edmund Tsui[20], Tamara S. Fraser[21], Benjamin Xu[22]

1 Population, Policy and Practice Department, UCL GOS Institute of Child Health, London, United Kingdom, 2 Great Ormond Street Hospital, London, United Kingdom, 3 Great Ormond Street NIHR Biomedical Research Centre, London, United Kingdom, 4 Cornea and External Eye Disease Service and Refractive Service, Singapore National Eye Center, Singapore, Singapore, 5 DUKE-NUS Department of Ophthalmology and Visual Sciences, Singapore, Singapore, 6 University Hospitals Sussex NHS Foundation Trust, London, United Kingdom, 7 Moorfields NIHR Biomedical Research Centre, London, United Kingdom, 8 UCL Institute of Ophthalmology, London, United Kingdom, 9 NIHR Birmingham Biomedical Research Centre, Birmingham, United Kingdom, 10 College of Medicine and Health, University of Birmingham, Birmingham, United Kingdom, 11 Department of Optometry and Vision Sciences, The University of Melbourne, Parkville, Victoria, Australia, 12 Department of Ophthalmology, Sheffield Teaching Hospitals NHS Foundation Trust, Sheffield, United Kingdom, 13 Trinidad Eye Hospital, San Fernando, Trinidad and Tobago, 14 Hamilton Glaucoma Center, The Viterbi Family Department of Ophthalmology, Shiley Eye Institute, University of California, San Diego, California, United States of America, 15 Birmingham Health Partners Centre for Regulatory Science and Innovation, Birmingham, United Kingdom, 16 Singapore Eye Research Institute, Singapore, Singapore, 17 Department of Optometry & Visual Sciences, City St George's, University of London, London, United Kingdom, 18 University College London, UCL Institute of Neurology, London, United Kingdom, 19 Departments of Ophthalmology and Neurology, Amsterdam UMC, the Netherlands, 20 Ocular Inflammatory Disease Center, UCLA Stein Eye Institute, David Geffen School of Medicine at UCLA, Los Angeles, California, United States of America, 21 Mid South Essex NHS Foundation Trust, Essex, United Kingdom, 22 USC Roski Eye Institute, Keck School of Medicine at USC, Los Angeles, California, United States of America

* a.solebo@ucl.ac.uk

## Abstract

### Background

Anterior segment optical coherence tomography (AS-OCT) is emerging as a valuable diagnostic, monitoring and predictive tool. Clinical utility has been suggested for ophthalmic disorders such as glaucoma, corneal disease, cataract and uveitis, which taken together comprise the majority of the blinding conditions affecting working age individuals globally. anterior segment is an obstacle to reproducibility and interoperability. To provide this guidance, we aim to extend the existing Advised Protocol for OCT Study Terminology and Elements (APOSTEL) guidelines, to ensure applicability to AS-OCT.

**Data availability statement:** Deidentified research data generated by the study described in this protocol will be made publicly available when the study is completed and published.

**Funding:** AL Solebo is supported by an NIHR Clinician Scientist grant (https://www.nihr.ac.uk CS-2018-18-ST2-005). All research at UCL Great Ormond Street Institute of Child Health is made possible by the NIHR Great Ormond Street Hospital Biomedical Research Centre. Prof Keane is supported by a UK Research & Innovation Future Leaders Fellowship (https://www.ukri.org/what-we-do/developing-peo-ple-and-skills/future-leaders-fellowships/, MR/T019050/1) and The Rubin Foundation Charitable Trust (https://findthatcharity.uk/orgid/GB-CHC-327062) The funders had no role in study design, data collection and analysis, decision to publish, or preparation of the manuscript.

**Competing interests:** Funding information AL Solebo is supported by a National Institute for Health and Care Research (NIHR) Clinician Scientist grant (https://www.nihr.ac.uk, grant reference CS-2018-18-ST2-005), and a Wellcome Clinician Scientist award (Grant funding schemes and guidance | Research funding | Wellcome, grant reference 311252/Z/24/Z). All research at UCL Great Ormond Street Institute of Child Health is made possible by the NIHR Great Ormond Street Hospital Biomedical Research Centre. Prof Keane is supported by a UK Research & Innovation Future Leaders Fellowship (https://www.ukri.org/what-we-do/developing-people-and-skills/future-leaders-fellowships/, grant reference MR/T019050/1) and The Rubin Foundation Charitable Trust (https://findthatcharity.uk/orgid/GB-CHC-327062, no grant reference) The funders had no role in study design, data collection and analysis, decision to publish, or preparation of the manuscript. Competing interests I have read the journal's policy and the authors of this manuscript have the following competing interests: Dr Keane has acted as a consultant for Retina Consultants of America, Topcon, Roche, Boehringer-Ingleheim, and Bitfount and is an equity owner in Big Picture Medical. He has received speaker fees from Zeiss, Novartis, Gyroscope, Boehringer-Ingleheim, Apellis, Roche, Abbvie, Topcon, and Hakim Group. He has received travel support from Bayer, Topcon,

## Methods

In line with EQUATOR Network guidance for the development of reporting guide-lines, APOSTEL-AS will be developed through a staged consensus process involv-ing literature review and Delphi consensus across an international multi-disciplinary stakeholder group, overseen by a multi-disciplinary multi-national Steering Commit-tee. The systematic scoping review will be used to generate candidate items, sup-port the development of a consensus nomenclature for AS-OCT representation of ocular structure, and to form Delphi group membership. Delphi methodology, used to consider items for inclusion, rewording or exclusion, will be undertaken in line with ACCORD (ACcurate COnsensus Reporting Document) guidance, with at least two rounds of Delphi survey, inclusion consensus threshold set at 80%, and steer-ing committee reviews between rounds. The resultant APOSTEL-AS guideline will undergo piloting before dissemination of the final version.

## Discussion

The APOSTEL-AS checklist, with minimum and recommended items to be reported about study methods, should provide timely support for researchers to ensure future standardisation, interoperability and reproducibility of reported work, hastening imple-mentation and the translation of knowledge into clinically beneficial action.

## Introduction

Optical coherence tomography (OCT) has changed the face of ophthalmic clinical practice and medical research. The research and clinical utility of ocular OCT bio-markers has been demonstrated within ophthalmology [1–3], and beyond the eye [4]. Quantitative data from posterior segment ocular OCT, such as retinal nerve fibre layer or central macular thickness, have been used as diagnostic, predictive, prognostic and monitoring biomarkers for inflammatory, degenerative, vascular, and metabolic disorders [4–7].

The validation and adoption of these biomarkers has been enabled by the standardisation and harmonisation of the reporting of research findings within the underpinning evidence base. The 2016 and 2021 Advised Protocol for OCT Study Terminology and Elements (APOSTEL) and APOSTEL2.0 recommendations [8,9] for reporting quantitative retinal OCT studies strengthened reproducibility and interop-erability across OCT studies of the posterior segment of the eye. However, some aspects of the current APOSTEL guidance are not applicable to anterior segment OCT (AS-OCT) studies. AS-OCT is being used to derive quantitative data from images of the normal physiology of the anterior ocular structure, data which carry clinical utility for the diagnosis and management of diseases involving those struc-tures. These diseases include tear film disorders, corneal pathology, glaucoma, cataract and anterior uveitis [10–16], disorders which, collectively, constitute the most common causes of working aged blindness globally [17].

and Roche. He has attended advisory boards for Topcon, Bayer, Boehringer-Ingleheim, RetinAI, and Novartis. Dr Solebo has acted as a consultant for Alimera Sciences and received speaker fees from Heidelberg Engineering Limited. No other author has competing interests.

In order to improve the reproducibility and interoperability of AS-OCT studies, we aim to develop a checklist of the core information that should be provided when reporting studies in which the quantitative assessment of AS OCT structures is undertaken. This checklist will be developed using the framework of the current APOSTEL recommendations [8].

## Materials and methods

APOSTEL-AS has been registered as a reporting guideline under development on the EQUATOR (Enhancing the QUality and Transparency Of health Research) library of reporting guidelines and will be developed in accordance with the EQUATOR Network's methodological framework. A steering committee with representation of expertise in tear film, corneal, lens, glaucoma and uveitic disorders, expertise in multinational, multi-disciplinary consensus exercises, and including investigators involved in developing the APOSTEL guidelines has been formed to oversee the conduct and methodology of the study. A multi-stage Delphi consensus approach will be used (Fig 1).

### Literature review and candidate item generation

An initial list of candidate items for the APOSTEL-AS checklist will be generated through review of the published literature and consultation with the steering committee (SC). A database search will be performed, with searches of the Medline, Embase and CINAHL databases from inception using terms including "anterior segment optical coherence tomography", "anterior eye segments" and "oct" (details provided within supplemental document, search approach) to identify eligible studies. Eligibility criteria comprise publications reporting the analysis of quantitative data derived from AS-OCT acquired from human eyes of living participants.

n and extracted as potential reporting items using a study-specific data collection form (DCF) *developed de novo* by the SC. Data extraction to the DCF will be undertaken independently by at least two investigators. Items will be reframed as candidate reporting items ahead of the consensus process. The SC will also create a consensus nomenclature for ocular anterior segment structures visible on axial cross sectional OCT imaging. This consensus nomenclature will be developed by the APOSTEL-AS SC using the nominal group technique, with consensus reached through discussion and majority voting.

Delphi consensus group membership:

Membership of the Delphi consensus group will be informed by the findings of the systematic review. Corresponding authors of studies reporting quantitative anterior segment OCT results will be contacted by email, requesting a response within six weeks. Reminder emails will be sent three weeks later. Records will be kept of attempted email and confirmed contact. Response rates will be reported using those who complete the Delphi as the numerator, and those for whom receipt of the email can be confirmed as the denominator. Other stakeholders to be identified and invited by the SC to join the Delphi consensus group comprise eye healthcare professionals,

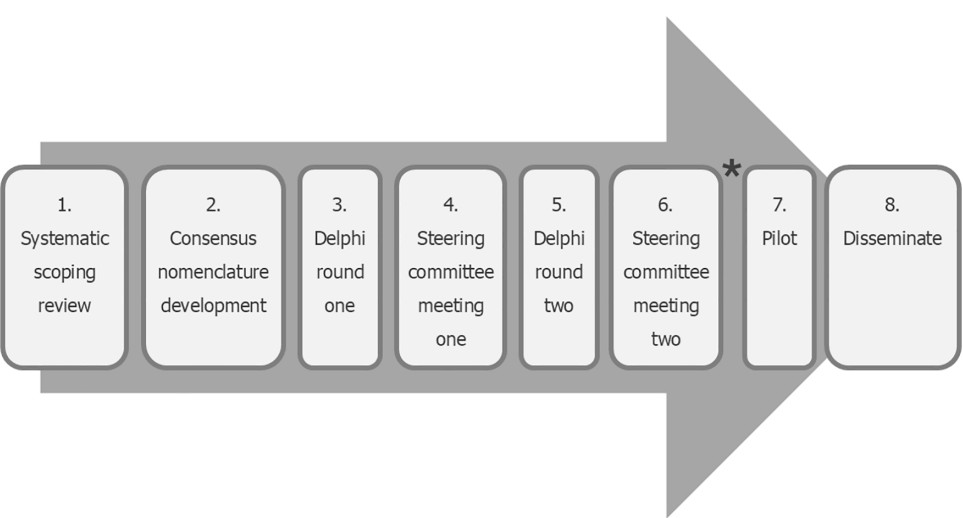

**Fig 1. APOSTEL-AS development process.** *Further Delphi rounds may be needed to reach consensus.

patients, methodologists, statisticians, computer scientists, industry representatives, health informaticists, and journal editors. Experts will be allowed to support snowball recruitment (i.e., to suggest additional experts). Records will be kept of all those successfully contacted, either directly by the SC or as corresponding authors, or indirectly through snowball recruitment.

Online Questionnaire 1 (Delphi 1):

Participants will be given written information about the study and asked to state their professional role, region of origin, and level of expertise (using years active). The candidate reporting items generated by the systematic review will be presented to the Delphi consensus group members, alongside the existing APOSTEL 2.0 items. Members will be asked to vote on each item using a 3-point scale: not important, important but not critical, and important and critical. They will also be asked about their agreement on the wording of each item and given the opportunity to submit free text comments about suggested amendments to wording of additional items, and invited to suggest additional items.

OQ2 (Expert Group virtual meeting):

We will then form a panel of experts comprising the SC and selected members of the Delphi consensus group (ensuring representation of stakeholder groups), who will convene at a virtual meeting. The aggregated results of the initial questionnaire, comments made by participants) will be reviewed by the panel. Following open discussion of the Delphi results, anonymous electronic voting will be used to include or exclude each reworded, amended, or newly suggested item. An 80% threshold will be used to demonstrate majority consensus for the inclusion of an item.

OQ3 (Delphi 2): The included items will be considered in a second Delphi round by the original Delphi consensus group. In this round, participants will be asked to approve or reject the list of APOSTEL-AS items. Items achieving consensus ≥80% will be included. Any item which does not achieve consensus but is approved by the majority (51–79% inclusive) will be reconsidered by the expert group for consideration following further amendment or refinement, and representation in a subsequent round (Delphi 3).

OQ4 (Delphi 3): Those items that reached majority consensus but did not achieve the 80% consensus threshold will be represented to the Delphi consensus group following amendment by the SC for one final Delphi round. Participants will again be asked to approve or reject the items. Again, items achieving consensus ≥80% will be included. Any item that

does not achieve consensus but is approved by the majority (51–79% inclusive) will be reconsidered within subsequent Delphi rounds (OQ5+) by the expert group for consideration for further amendment or refinement prior to inclusion in the penultimate list.

### Guideline pilot

The penultimate guideline will be checked by the Operations team (ALS, AB, AF, TT) to ensure its clarity. It will then be presented to at least five stakeholders for piloting. The stakeholders participating in this pilot will comprise Delphi consensus participants who were not part of the expert group, or additional external experts identified following the start of the Delphi process. The Operations team will use feedback from this pilot to refine the guideline and produce the final version of the checklist.

### Study status and timelines

Study stages up to OQ2 (expert group virtual meeting) have been completed (by date November 10th, 2024). Following completion of OQ3, OQ4 and piloting, the final tool is expected to be produced in early 2026.

Ethics approval for this work has been granted by the University College London Institutional Review Board (UCL Research Ethics Committee Approval ID Number: 26253/001). Informed electronic consent will be obtained from consensus group members.

## Discussion

The APOSTEL-AS recommendations will be an extension to the current APOSTEL guidelines, adding specific parameters for the reporting of AS-OCT data. AS-OCT is increasingly being used to derive quantitative data from images of normal physiology and for diseases involving the anterior segment of the eye. There are several differences in the acquisition and analysis of anterior versus posterior segment OCT images. Whilst posterior segment OCT (PS-OCT) concerns images of tissues with similar refractive indices and typically measuring no more than 1 mm in depth (as measured from the inner limiting membrane to the sclera), with the *'en face'* area of interest typically being $4mm^2$, the ocular structures imaged in AS-OCT are arranged over a greater depth (at least 10 mm, as measured from the corneal epithelium to the anterior hyaloid area) and with a much wider area of interest depending on the clinical or research need [18]. Furthermore, whilst the posterior segment tissues minimally scatter light, there are unique features related to light transmissibility of the different anterior ocular structures, from the optically clear cornea to the optically opaque sclera [18]. These considerations differentiate AS-OCT from PS-OCT, but it is also likely that these challenges result in heterogeneity of study methods, including acquisition processes, image quality assessment, data extraction and image analysis across the AS-OCT literature. APOSTEL-AS will provide timely support for users of AS-OCT, to ensure future standardisation, interoperability and reproducibility of reported research and clinical findings.

A limitation of the approach described in this protocol is the restriction of the identification of potential consensus group members to those researchers who have published eligible studies within the preceding four years. High impact, novel, innovative imaging studies may require lengthy recruitment periods, which may mean that our time-restricted approach does not involve scientists whose work is not yet disseminated through peer-reviewed publications. The use of 'snowball' extension (i.e., consensus group members identifying other potential members) should help to address this limitation. Strengths of the approach include the involvement of a wide range of stakeholder expertise and experience, across multiple countries, with collection of information about their degree of experience and expertise to inform the contextualisation of responses. Development of items will be underpinned by a comprehensive systematic review, with collection of data from manuscripts published over the last 20 years. It is acknowledged that this comprehensiveness may result in the collection of historical data from imaging platforms that are no longer in use, due to being superseded by more contemporary devices.

Several consensus methods exist for the capture and description of consensus across multi-disciplinary groups. The nominal group technique (NGT) was selected for the initial activities of the SC, in view of its structured sequence of events, principal of equal participation, and promotion of diversity in opinion, which should support the work of this smaller group [19]. NGT consensus exercises can be time consuming, particularly for large groups, but this process will benefit from facilitation undertaken by subject matter experts with experience in supporting consensus initiatives [20,21]. The selection of the Delphi method for the development of the checklist enables this activity to be undertaken by a large, geographically and scientifically diverse group with different degrees and spheres of experience and expertise. A recently published EQUATOR (Enhancing the QUality and Transparency Of health Research) guideline for reporting consensus-based methods in biomedical research and clinical practice (the ACcurate COnsensus Reporting Document, or ACCORD) will be used to report the study findings [19].

The original APOSTEL checklist was developed following concerns around the absence of consensus around quality control criteria for posterior segment OCT imaging [9]. Similar concerns are emerging for AS-OCT imaging, with recognition of this issue being a particular obstacle to the potentially transformative role of artificial intelligence (AI) approaches to image analysis and the development of imaging-based biomarkers [22,23]. AI approaches hold the promise of dramatically increasing the utility of clinical images, with deep learning (DL) based image analysis in particular proving non-inferior to expert human assessment [12,24,25]. Human analysis of the quantitative information that can be extracted from anterior segment structures is subjective, open to inter- and intraobserver variability, and time consuming [26]. DL image analysis methods would provide objective, repeatable, reliable OCT biomarkers, but these approaches require large (or harmonised), diverse, annotated datasets. The current absence of explicit and comprehensive reporting of AS-OCT study methods is an obstacle to comparison or collation of existing annotated imaging datasets. Through the wide dissemination and uptake of the APOTEL-AS guidance, enabled by ensuring accessibility in open access formats and translation into other languages as directed and supported by stakeholders, this obstacle should be addressed.

## Supporting information

**S1 File. Literature search strategy.**
(DOCX)

## Acknowledgments

We thank the Enhancing the Quality and Transparency of health research Network (EQUATOR Network) team for their support with study development.

## Author contributions

**Conceptualization:** Ameenat Lola Solebo.

**Data curation:** Ameenat Lola Solebo.

**Formal analysis:** Ameenat Lola Solebo.

**Funding acquisition:** Ameenat Lola Solebo.

**Investigation:** Ameenat Lola Solebo, Marcus Ang, Alice Bellchambers, Colin J Chu, Alastair K Denniston, Laura E Downie, Thomas Evans, Alexander S Fraser, Scott Hau, Alex S Huang, Pearse A. Keane, Xiaoxuan Liu, Jodhbir S Mehta, Giovanni Ometto, Axel Petzold, Edmund Tsui, Tamara S Fraser, Benjamin Xu.

**Methodology:** Ameenat Lola Solebo, Marcus Ang, Alice Bellchambers, Colin J Chu, Alastair K Denniston, Laura E Downie, Thomas Evans, Alexander S Fraser, Scott Hau, Alex S Huang, Pearse A. Keane, Xiaoxuan Liu, Jodhbir S Mehta, Giovanni Ometto, Axel Petzold, Edmund Tsui, Tamara S Fraser, Benjamin Xu.

**Project administration:** Ameenat Lola Solebo.

**Resources:** Ameenat Lola Solebo.

**Software:** Ameenat Lola Solebo.

**Supervision:** Ameenat Lola Solebo.

**Validation:** Ameenat Lola Solebo.

**Visualization:** Ameenat Lola Solebo.

**Writing – original draft:** Ameenat Lola Solebo, Laura E Downie.

**Writing – review & editing:** Ameenat Lola Solebo, Marcus Ang, Alice Bellchambers, Colin J Chu, Alastair K Denniston, Thomas Evans, Alexander S Fraser, Scott Hau, Alex S Huang, Pearse A. Keane, Xiaoxuan Liu, Jodhbir S Mehta, Giovanni Ometto, Axel Petzold, Edmund Tsui, Tamara S Fraser, Benjamin Xu.

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
