## [Decision Letter · Decision Letter 0]

30 Jun 2025

Dear Dr. Solebo,

Thank you for submitting your manuscript to PLOS ONE. After careful consideration, we feel that it has merit but does not fully meet PLOS ONE’s publication criteria as it currently stands. Therefore, we invite you to submit a revised version of the manuscript that addresses the points raised during the review process. The reviewers comments are mostly favorable. However there are minor modifications which are required. 

We look forward to receiving your revised manuscript.

Kind regards,

Kumar Saurabh

Academic Editor

PLOS ONE

Journal Requirements:

2. Please expand the acronym “NIHR” (as indicated in your financial disclosure) so that it states the name of your funders in full.

5. Thank you for stating the following in the Competing Interests/Financial Disclosure section:

I have read the journal's policy and the authors of this manuscript have the following competing interests: Dr Keane has acted as a consultant for Retina Consultants of America, Topcon, Roche, Boehringer-Ingleheim, and Bitfount and is an equity owner in Big Picture Medical. He has received speaker fees from Zeiss, Novartis, Gyroscope, Boehringer-Ingleheim, Apellis, Roche, Abbvie, Topcon, and Hakim Group. He has received travel support from Bayer, Topcon, and Roche. He has attended advisory boards for Topcon, Bayer, Boehringer-Ingleheim, RetinAI, and Novartis. No other author has competing interests.

We note that one or more of the authors have an affiliation to the commercial funders of this research study : Retina Consultants of America, Topcon, Roche, Boehringer-Ingleheim, Bitfount

Additional Editor Comments:

Dear Authors

The reviewers comments are mostly favorable. However there are minor modifications which are required. I invite you to address the concerns mentioned in reviewers comments

Thank you

Reviewers' comments:

Reviewer's Responses to Questions

**Comments to the Author**

1. Does the manuscript provide a valid rationale for the proposed study, with clearly identified and justified research questions?

Reviewer #1: Yes

Reviewer #2: Yes

2. Is the protocol technically sound and planned in a manner that will lead to a meaningful outcome and allow testing the stated hypotheses?

Reviewer #1: Yes

Reviewer #2: Yes

3. Is the methodology feasible and described in sufficient detail to allow the work to be replicable?

Reviewer #1: Yes

Reviewer #2: Yes

4. Have the authors described where all data underlying the findings will be made available when the study is complete?

Reviewer #1: Yes

Reviewer #2: Yes

5. Is the manuscript presented in an intelligible fashion and written in standard English?

Reviewer #1: Yes

Reviewer #2: Yes

You may also provide optional suggestions and comments to authors that they might find helpful in planning their study.

Reviewer #1: Congratulations on attempting this important project that will help define - and standardize - terminology and elements in anterior segment OCT imaging.

Some suggestions that may be helpful for authors...

The "snowball extension" concept will be critically important to include 'persons of interest' who who have expertise but may not have published in past 4 years - especially because the research output has been lower due to worldwide pandemic.

Furthermore, it is important to contact and attempt re-contact the corresponding authors as suggested in your study because things can get "lost' in the email deluge that we all are accustomed to.

Consensus opinions will have greater acceptance and adoption with inclusivity of all the expert players.

Reviewer #2: 1.What is the motivation for this publication? This is the study protocol and one would normally want to publish the results of the study rather than the study protocol.

2.The methodology looks very appropriate for the main objective of the study but correct the highlighted errors as indicated in the attached copy of the manuscript.

**Do you want your identity to be public for this peer review?** For information about this choice, including consent withdrawal, please see our Privacy Policy

Reviewer #1: No

Reviewer #2: **Yes: ** Dr. Kwadwo Amoah

---

## [Author Response · Author response to Decision Letter 1]

12 Aug 2025

Author response: We have followed the provided file naming guidance for figures and supplemental data, and ensured our manuscript formatting with the PLOS ONE template provided

2. Please expand the acronym “NIHR” (as indicated in your financial disclosure) so that it states the name of your funders in full.

Author response: This has been done

Author response: We have ensured that grant references are labelled as such (as one of the reference numbers in this statement relates to the funding organisation, the Rubin Foundation, rather than a specific grant)

Author response: This has been done - we have stated that "All data are in the manuscript and/or supporting information files".

5. Thank you for stating the following in the Competing Interests/Financial Disclosure section:

I have read the journal's policy and the authors of this manuscript have the following competing interests: Dr Keane has acted as a consultant for Retina Consultants of America, Topcon, Roche, Boehringer-Ingleheim, and Bitfount and is an equity owner in Big Picture Medical. He has received speaker fees from Zeiss, Novartis, Gyroscope, Boehringer-Ingleheim, Apellis, Roche, Abbvie, Topcon, and Hakim Group. He has received travel support from Bayer, Topcon, and Roche. He has attended advisory boards for Topcon, Bayer, Boehringer-Ingleheim, RetinAI, and Novartis. No other author has competing interests.

We note that one or more of the authors have an affiliation to the commercial funders of this research study : Retina Consultants of America, Topcon, Roche, Boehringer-Ingleheim, Bitfount

Author response: This study has not been funded by any commercial organisations – we have now separated the funder details from the competing interests details to make this clear.

Author response: We have ensured that it is clear that there was no commercial funding for the study, and that the non-commercial funders had no role in the study

Author response: we have now separated the funder details from the competing interests details

Author response: We confirm that the reference list is complete and correct

Reviewer #1:

The "snowball extension" concept will be critically important to include 'persons of interest' who have expertise but may not have published in past 4 years - especially because the research output has been lower due to worldwide pandemic.

Furthermore, it is important to contact and attempt re-contact the corresponding authors as suggested in your study because things can get "lost' in the email deluge that we all are accustomed to.

Consensus opinions will have greater acceptance and adoption with inclusivity of all the expert players.

Author response: We thank the reviewer for their support for our methods (use of snowball recruitment as described in line 152, use of reminder emails as described in line 149, and the consensus approach). We note that no change has been suggested

Reviewer #2: 1.What is the motivation for this publication? This is the study protocol and one would normally want to publish the results of the study rather than the study protocol.

Author response: We thank the reviewer for asking us to explicitly state that the purpose of publishing the protocol is / was* to take an ‘Open Science’ approach to support and to support engagement for our activities across the stakeholder community of researchers and clinicians.

(*We have amended timelines within the article as the protocol was originally submitted some time ago.)

2.The methodology looks very appropriate for the main objective of the study but correct the highlighted errors as indicated in the attached copy of the manuscript.

Line 114 This should be 'quantitative'

Author response: This has been amended, thank you

Line 134 Elaborate this a little bit more. A busy reader may not have the time to refer to the supplemental document.

Author response: More details have now been provided

Line 136, “Specific considerations for Screening, selection AS-OCT”: This doesn't sound meaningful. Can you rephrase it?

Author response: Changed to “Specific considerations for acquisition, selection, processing and analyses of AS-OCT will be identified from these studies”, thank you for noting this error

Line 157 this should be 'numerator'

Author response: This has been amended, thank you

---

## [Editor Report · Decision Letter 1]

14 Aug 2025

Development of the Advised Protocol for OCT Study Terminology and Elements Anterior Segment OCT extension reporting guidelines (APOSTEL-AS): Study protocol

PONE-D-24-47643R1

Dear Dr. Solebo,

We’re pleased to inform you that your manuscript has been judged scientifically suitable for publication and will be formally accepted for publication once it meets all outstanding technical requirements.

Kind regards,

Kumar Saurabh

Academic Editor

PLOS ONE
---

## [Editor Report · Acceptance letter]

PONE-D-24-47643R1

PLOS ONE

Dear Dr. Solebo,

I'm pleased to inform you that your manuscript has been deemed suitable for publication in PLOS ONE. Congratulations! Your manuscript is now being handed over to our production team.

Kind regards,

on behalf of

Dr. Kumar Saurabh

Academic Editor

PLOS ONE